# Occupational Exposure Assessment to Antineoplastic Drugs in Nine Italian Hospital Centers over a 5-Year Survey Program

**DOI:** 10.3390/ijerph19148601

**Published:** 2022-07-14

**Authors:** Cristina Sottani, Elena Grignani, Marco Cornacchia, Sara Negri, Francesco Saverio Robustelli della Cuna, Danilo Cottica, Dario Bruzzese, Paolo Severi, Daniele Strocchi, Giovanni Verna, Veruscka Leso, Ivo Iavicoli

**Affiliations:** 1Environmental Research Center, Istituti Clinici Scientifici Maugeri IRCCS, 27100 Pavia, Italy; elena.grignani@icsmaugeri.it (E.G.); sara.negri@icsmaugeri.it (S.N.); saverio.robustelli@icsmaugeri.it (F.S.R.d.C.); danilo.cottica@icsmaugeri.it (D.C.); 2Deparment of Drug Science, University of Pavia, 27100 Pavia, Italy; marco.cornacchia01@universitadipavia.it; 3Department of Public Health, Section of Medical Statistics, University of Naples Federico II, 80131 Naples, Italy; dario.bruzzese@unina.it; 4Prevention and Protention Service, Azienda Unità, Sanitaria Locale della Romagna Ospedali degli ambiti territoriali di Ravenna e Rimini dell’AUSL della Romagna, 47521 Cesena, Italy; paolo.severi@auslromagna.it (P.S.); daniele.strocchi@auslromagna.it (D.S.); giovanni.verna@auslromagna.it (G.V.); 5Department of Public Health, Section of Occupational Medicine, University of Naples Federico II, 80131 Naples, Italy; veruscka.leso@unina.it (V.L.); ivo.iavicoli@unina.it (I.I.)

**Keywords:** chemotherapeutic drugs, frequency of positive samples, percentiles of data distribution, temporal trends, threshold exposure limits

## Abstract

In the present study, surface contamination where antineoplastic drugs (ADs) are present was investigated, as occupational exposure risk is still an open debate. Despite recommendations and safety standard procedures being in place in health care settings, quantifiable levels of ADs are being reported in the recent literature. Thus, a survey monitoring program was conducted over five years (2016–2021) in nine Italian hospitals. The repeated surveys produced 8288 data points that have been grouped according to the main hospital settings, such as pharmacy areas and patient care units. Based on the most often prepared ADs, the investigated drugs were cyclophosphamide (CP), gemcitabine (GEM), 5-fluorouracil (5–FU), and platinum compounds (Pt). Patient care units had a frequency of positive wipe samples (59%) higher than pharmacies (44%). Conversely, pharmacies had a frequency of positive pad samples higher (24%) than patient care units (10%). Moreover, by statistical analysis, pad samples had a significantly higher risk of contamination in pharmacy areas than in patient care units. In this study, the 75th and the 90th percentiles of the contamination levels were obtained. The 90th percentile was chosen to describe a suitable benchmark that compares results obtained by the present research with those previously reported in the literature. Based upon surface contamination loads, our data showed that 5–FU had the highest concentration values, but the lowest frequency of positive samples. In pharmacy areas, the 90th percentile of 5–FU data distribution was less than 0.346 ng/cm^2^ and less than 0.443 ng/cm^2^ in patient care units. AD levels are higher than those reported for health care settings in other European countries yet trends of contamination in Italy have shown to decrease over time.

## 1. Introduction

The last updated estimates by GLOBOCAN 2020 confirmed the continuous increase of cancer incidences worldwide as 18 million new cancer cases were diagnosed in 2018 and 19.3 million in 2020 [1]. This landscape is consistent with a consequent and increasing use of antineoplastic drugs (ADs) in health care centers to treat cancer. The list of ADs and other hazardous drugs have regularly been updated by the U.S. National Institute for Occupational Safety and Health (NIOSH) since 2004 [2]. According to recent surveys, cyclophosphamide (CP) (classified as carcinogenic to humans by the International Research on Cancer, IARC), ifosfamide (IF), methotrexate (MTX), gemcitabine (GEM), 5–fluorouracil (5–FU), irinotecan (IRT), etoposide (ETP), taxanes (TX), and vinorelbine (VNR) are the most used ADs [3,4,5,6]. The literature has seen a number of papers being published with a focus on several issues regarding the topic of occupational exposure to ADs [7,8,9,10,11,12,13]. As a matter of concern, to evaluate the entire workplace and to define a proper protection level for hospital personnel, measures to limit the spread of contamination are continuously taken by practitioners. Today, although new technologies and/or work processes and organizational and technical measures are implemented, and personal protective equipment (PPE) such as gloves, gowns, respiratory, and eye protection is used, the risk of exposure to residual concentrations of drugs is still evident [3,4,14]. On the other hand, the only certain point is that ways of reducing occupational risk exists and now reducing it as low as reasonably achievable (ALARA) remains the commonly shared objective among the community of occupational health professionals. For ADs, no occupational exposure limits have been currently established by national or international organizations. Kiffmeyer et al. [15] provided a substance independent guideline based on the 90th percentile for eight ADs (CP, IP, 5–FU, GEM, IRT, ETP, MTX, and TX). The 90th percentile proposed by this study was 0.1 ng/cm^2^. In addition, our group recommended technical threshold limits for four ADs (CP, 5–FU, GEM, and platinum-coordinated compounds) based on the 90th percentile of wipe sampling data distribution. Environmental results from monitoring programs were obtained between 2009 and 2011 [16]. Specific threshold limits for each drug were suggested as these antineoplastic agents were grouped according to their different toxicological profile and the classification of IARC [6].

In the literature, these limits based on a traffic-light model were also suggested [3,17,18,19]. In the final ‘traffic-light’ model, Sessink suggested a stepwise approach. For CP, monitoring surveys are required once a year with levels less than 0.1 ng/cm^2^ and repeated monitoring surveys (i.e., within 3–6 months) are requested with levels between 0.1 and 1.0 ng/cm^2^. Moreover, urine sampling remains a permanent tool when concomitant environmental monitoring programs are carried out [17]. In other studies, the 75th percentile along with the 90th percentile were reported for each drug, and they were based on surface contamination loads found in both pharmacies and patient care units [20,21,22,23]. The use of the 90th percentile of data distribution was considered as a good indicator to define not only the contamination level, but also a significant variability over time.

In Italy, instead, to the best of our knowledge, no investigations to define surface contamination thresholds have been recently developed according to the lower limit of detection values (LODs) of the most recently validated analytical methods.

In our study, a survey program was carried out according to Italian guidelines in hospitals located in one region of Italy where the same policies for the safe handling of ADs were set out in 2012 [24]. Therefore, in order to minimize discrepancies in the management of the risk for exposure to ADs, both pharmacies and patient care units were chosen as belonging to the same group of health care settings. The repeated surveys over time (2016–2021) produced 8288 data points that have been grouped according to the main hospital settings (compounding and administration) on a temporal trend basis. Moreover, the impact of decontamination/deactivation solutions on the presence of residues still persistent in these workplaces was evaluated. The aims of the present study were: to determine and compare the levels of contamination in both pharmacies and patient care units, to provide the frequency of positive measurements in workplaces, and to assess threshold limits (the 90th percentiles) for ADs for checking if the exposure is adequately controlled.

## 2. Materials and Methods

### 2.1. Study Design

The study was conducted in a region of northern Italy and 9 hospital centers were selected for the study. They were numbered from 1 to 9. Table 1 summarizes key center characteristics. Based on the number of beds, the hospitals were classified as small (<50 beds), medium (50–150 beds), and large (>150 beds). Hospitals 1, 2, and 6 include both specialized pharmacies, equipped with laminar airflow biosafety cabinets (BSCs) for sterile compounding, and patient care units for drug administration (including nursing and patient areas). The pharmacy of the hospital 6 is equipped for both running manual and robot-assisted compounding (APOTECAchemo^®^). In order to make a comparison, all results related to APOTECA (Ancona, Italy, Loccioni Humancare)were excluded from the present study.

At the end of the preparation step, chemotherapy-infusion bags are packed under the hood and placed in a plastic bag on trays outside the hood by the pharmacy technician. Then, the preparation is transported from the pharmacy areas to the patient care units in a box intended for hazardous drugs.

To evaluate potential exposure to CP, 5–FU, GEM, and Pt, wipe samples were collected together with pad samples. For the hospitals numbered as 1, 2, and 6, a total number of 319, 433, and 297 wipe samples were taken, respectively. For the hospitals numbered as 3, 4, 5, 7, 8, and 9, surface wipe sampling included the collection of 150, 60, 120, 60, 136, and 122 samples, respectively. Two work shifts, such as the start of a shift in the morning and the end of a shift in the evening (before the room was cleaned) were studied.

AD exposure was evaluated by using pad samples and a total number of 375 samples were collected over the 5-year monitoring study. A hundred and twenty-five workers wore pad tests in correspondence with left and right forearms and the thorax. The schematic presentation of the study design is reported in Figure 1. It displays both the type and the number of samples collected over time and how workers were distributed by each hospital site.

Each participating hospital was expected to apply the same local policies and procedures for compounding, administration, and surface cleaning. When the study was complete, each center received a report comparing its results with global findings from all participating centers. The results were, therefore, disseminated among the participating staff.

### 2.2. Sampling Techniques

The survey was conducted once a year with both pharmacy and patient care units. Surface wipe sampling was used to determine workplace contamination with ADs. Wipe sampling was performed according to a procedure previously described by our group [25]. For each sampling location, a standardized area of 20 × 20 cm (i.e., 400 cm^2^) was sampled with nonwoven gauze (TNT Type Luxor-Net, STS Medical Group, Luigi Salvadori S.p.A., Scandicci, Italy). However, for all surfaces or objects for which this size was not applicable, the sampled area was exactly measured and then recorded. To clean the surface area, we developed an easy onsite wiping procedure. Instead of using tape, a permanent marker was used to mark the sampling area at the four corners. After the surface sampling, the gauze was folded and introduced into a 20 mL needle-free polypropylene syringe and closed inside the syringe using the piston. To avoid possible crosscontamination, the nitrile medical gloves worn by the operator were changed at each wipe-test. On the sampling day, the samples were stored in a fridge bag. In the laboratory where the analyses were performed, the samples were stored at −20 °C. Then, each wipe was moistened, at the time of sampling, with 2.5 mL of formic acid 0.1% (methanol for 5–FU). All the surfaces and objects were swept clean using vertical and horizontal strokes in two different directions (up and down, right and left). All the locations were sampled twice per monitoring day by using the same process; at the start of the morning shift, after surfaces and rooms were cleaned and, at the end of the evening shift, before the surfaces were cleaned and rooms were washed.

Moreover, to evaluate potential skin exposure of ADs, pad samples were used. A nonwoven gauze of 10 × 10 cm (i.e., 100 cm^2^) was worn by the hospital personnel during a normal day of work. The upper part of the body was thus monitored. Hospital personnel involved in the study put three pad samples on their gowns on the left and right forearms and the thorax.

### 2.3. Analytical Methods

The wipe desorption was performed by flowing three aliquots of 4.5 mL formic acid 0.1% in water through the syringe. The total eluate was briefly stirred and centrifuged for 3 min at 5000 rpm, thereby the sedimentation of any particulate matter was allowed. One ml aliquot was transferred into a 1.5 mL vial of the UPLC autosampler. A volume of 7.5 μL of sample was injected into an Acquity UPLC HSS T3 column (1.8 μm, 2.1 × 50 mm, Waters, Milan, Italy) at 35 °C. The mobile phases were: water (A1) and acetonitrile (B1), 99/1, and the flow rate was 0.45 mL/min. The analysis time was 5 min and the complete analytical cycle was 9 min. The lower detection limits of the method were: 0.1 ng for CP, GEM, Pt, and 5 ng for 5–FU. The lower quantification limits were: 0.5 ng for CP, GEM, Pt, and 10 ng for 5–FU [25]. Pt was measured by inductively coupled plasma mass spectrometry using a Perkin–Elmer Sciex^®^ ELAN 5000 ICP-mass spectrometer (Concorde, Ontario, Canada) equipped with an ultrasonic nebulizer, Cetac Technologies mod. U–500 [26]. In this study, LOD values were used for wipe and pad data analysis when the results were less than the lower limit of detection.

### 2.4. Statistical Analysis

For wipe tests, a sample was considered positive if at least one drug had a value above the LOD. The drug level was reported as a normalized value and expressed as ng/cm^2^. Most sampled locations covered an area of 400 cm^2^. Measurements below the LOD were instead imputed at their normalized LODs. The 75th and 90th percentiles of drug concentrations were calculated if at least 25% or 10% of drug measurements, respectively, were above the LODs. The description of each location sampled at both pharmacy and patient care areas is detailed in the following paragraphs. Chi-square test was used to explore the association between the presence of positive samples and the sampling technique (wipes/pads). This association was further explored by computing the odds ratios (ORs) with the corresponding 95% confidence intervals (95% CIs) that measure the increase (decrease) in the odds of a positive sample in pads with respect to wipes. ORs > 1 denote greater odds (i.e., risk) of a positive sample in pads with respect to wipes while ORs < 1 refer to greater odds of a positive sample in wipes with respect to pads. *p* values < 0.05 were deemed statistically significant.

## 3. Results

A survey monitoring program was conducted over six years (2016–2021). No monitoring program was carried out in 2018 because three pharmacies (1, 2, and 6) were being renovated in 2018. The most used drugs were 5–FU (median 12400 (2481–63,534) g/years), GEM (6010 (1650–29,062) g/years), CP (2560 (0.568–6214) g/years), and Pt (1600 (0–2892) g/years). For the surface contamination assessment, wipe tests were used. Potential dermal exposure of the hospital personnel was carried out by means of pad tests. In pharmacies, 489 wipe samples and 90 pad samples were taken from 30 technicians. In patient care areas, 1208 wipe samples and 285 pad samples were obtained from 95 nurses.

### 3.1. Frequency of Positive Data

#### 3.1.1. Wipe Samples

Drug measurements (6788) obtained by analyzing wipe samples were collected over a 5-year survey, and 55% (3721) were above the LOD. Surprisingly, the proportion of wipe samples that was positive for at least one antineoplastic drug was greater in the patient care units than in the pharmacies (Table 2). Fifty-nine percent of AD measurements in the administering areas vs. forty-four percent in the pharmacies were above the LOD. In particular, AD measurements above the LOD had a mean value of 32% until 2019. The frequency of AD measurements almost doubled between 2020 and 2021 (62%). In the outpatient sites, the frequency of positive samples remained similar over the years (Table 2).

#### 3.1.2. Pad Samples

Of the 1500 drug measurements taken, 207 (14%) were above the LOD.

Pharmacies had a higher frequency (24%) of pad samples with at least one drug above the LOD compared to administration areas (10%), as shown in Table 3.

Trends of variability over the 5-year survey period mirrored the percentage values reported above.

### 3.2. Pharmacy Areas

#### 3.2.1. Wipe Samples

Table 4 shows the wipe sample results in relation to the sampling locations. The 75th percentile, the 90th percentile, and the maximum values for concentration of contamination (ng/cm^2^) for the three (1, 2, and 6) hospital sites (pharmacy units) are reported. Fifteen locations were chosen following the criteria of those frequently touched as well as regularly reported in the previous literature [22,27]. The most contaminated locations were barcode surfaces for checking pharmacy documentation (62%), floors, particularly, those in front of BSC (56%), trays for drug delivery (51%), worktop surfaces inside BSC (38%), and handles, such as pass-through chamber handles (35%). Moreover, for both compounding and administering areas, the risk of contamination in pad samples vs. wipe samples was calculated. In pharmacies, 5–FU had odds ratio > 1 over the 5-year survey. A detailed table in this regard, including percentages of positive samples, odds ratio, and *p* values for wipe and pad samples is reported (Appendix A).

As the 75th percentile shows 25 percent of positive samples, the 90th percentile was chosen to describe a suitable benchmark that compares results obtained by the present study with those previously reported in literature. As an example, GEM and CP had the highest frequency of contamination as they were detected on more than 50% of the sampled surfaces in pharmacies, in contrast, they showed the lowest concentration values with the overall 90th percentile levels of 0.028 and 0.026 ng/cm^2^, respectively (Table 4). On the other hand, 5–FU showed the lowest prevalence of measurements above the LOD (28%) with the highest values of concentration. Figure 2, panels A and B, describes the concentrations of each drug over time.

The maximum values of 5–FU were obtained in 2017 and 2021. The 5–FU levels were 6.307 ng/cm^2^ and 9.270 ng/cm^2^ in correspondence with BSC work surfaces and, the 90th percentile of data distribution was 1.375 ng/cm^2^ (Table 4). The floor in front of BSC was also found to be contaminated by 5–FU (5.974 ng/cm^2^) in 2021. GEM was the drug with the second highest concentration. A value of 2.700 ng/cm^2^ of GEM was detected on the pass-through chamber handle inside the cleanroom in 2017. In pharmacy areas, other surfaces such as door handles showed quantifiable concentrations with values ranging from 0.085 to 0.118 ng/cm^2^ and the trays used for drug delivery showed values from 0.013 to 1.735 ng/cm^2^.

Furthermore, wipe test measurements were obtained after the completion of cleaning procedures and at the end of drug compounding. These values, normalized to percent values, from 2016 through 2021, were of roughly the same order of magnitude but doubled between 2020 and 2021, as shown in Figure 3, panel A. Patient care units such as pharmacy areas had AD measurements after the completion of cleaning procedures and at the end of drug compounding. These values, normalized to percent values, from 2016 through 2021, were of roughly the same order of magnitude and remained similar over the years of the study, as shown in Figure 3, panel B.

As regards the cleaning procedure from surfaces, AD residues were detected at different locations when using a solution of 3 × 10^−2^ M sodium dodecyl sulfate/isopropanol 80:20, v/v (homemade sterile solution). As an example, a concentration of 5–FU (1.409 ng/cm^2^) was found on the BSC worktop in 2017. Thus, a different mixture and composition of cleaning/decontamination solutions was used. The concentration of 5–FU at the same location was found to change from 1.409 ng/cm^2^ to 0.022 ng/cm^2^ at the end of the cleaning procedures when using 5% sodium hypochlorite. This data was reported in 2021. Sampling locations in cleanrooms, such as floors, handles, and trays used for drug delivery, conversely showed AD residues that were persistent even after the completion of the cleaning procedures.

#### 3.2.2. Pad Samples

The frequency of positive pad samples was 24% in pharmacy areas and 10% in patient care units (Table 2). For 5–FU, the highest value (46.067 ng/cm^2^) was recorded in 2016. This data was obtained by analyzing the sample put on the left forearm of a technician working at the BSC. The mean concentration was 0.252 ng/cm^2^ over the years of the study. In fact, the mean values ranged from the highest one (0.725 ng/cm^2^) in 2016 to the lowest level of 0.044 ng/cm^2^ in 2021. 

### 3.3. Patient Care Units

#### 3.3.1. Wipe Samples

The 75th percentile, the 90th percentile, and the maximum values for concentration of contamination (ng/cm^2^) at the nine hospital sites are reported in Table 5. Twenty-one locations were chosen at both inpatient bed wards and outpatient units where the survey monitoring programs were carried out.

A total of 4832 AD measurements were obtained, out of which 2853 (59%) were above the LOD. GEM was the most detected drug (77%), followed by Pt (68%), CP (64%), and 5–FU (33%), as shown in Table 2.

More in detail, the most contaminated locations were floors (73%) with 1828 AD measurements, particularly contaminated were those in front of the pole for the patient infusion bags (78%), followed by floors in front of restrooms (74%). Other locations with a high load of ADs were those inside the patient rooms (71%), touchscreens of the perfusion pumps (74%), armrests of the patient treatment chairs (72%), and counters used for deposition of ready-to-use bags (46%).

The 90th percentile of 5–FU had the maximum value of 1.464 ng/cm^2^ in correspondence with the location named as the pole for the infusion bags (Table 5). The concentration of this drug had significant values over the 5-year survey program. The amount of 5–FU detected at the pole for infusion bags had values of 7.023, 4.753, 0.810, 4.979, and 5.560 ng/cm^2^ per each year of the monitoring survey (2016–2021) (data not shown).

This set of data is of pivotal importance for outlining which location needs a regular monitoring assessment over time. The drug 5–FU had, therefore, a low prevalence of positive wipe samples but obtained the highest values of concentrations. CP, in contrast, showed the lowest concentration values with the overall 90th percentile level of 0.070 ng/cm^2^, as shown in Table 5.

In patient care areas, the contamination from 5–FU was present at different locations. This drug was detected on the armrest of the patient treatment chair with a value of 96.315 ng/cm^2^ in 2019. In correspondence with the tray used for delivering bags to patients, 5–FU showed a value of 7.403 ng/cm^2^. These trays are generally used by nurses for drug transportation. For 5–FU, the maximum value (112.500 ng/cm^2^) occurred in 2020. The location was the counter used for the deposition of ready-to-use bags at the nurse station. The 90th percentile of 5–FU data distribution (Table 5) confirmed that the contamination level found at the nurse station had the highest value of 6.666 ng/cm^2^. Additionally, 5–FU was the most prepared drug with a median amount of 12,400 g/years and the most administered one, as well.

As regards the cleaning procedure of ADs from surfaces, the amount of drugs found at the end of the cleaning procedures was often higher than measurements obtained at the end of work shifts for the personnel involved in activities of drug administration. Regarding trays for drug delivery, for example, the prevalence of positive measurements of 5–FU was equally distributed between the end of the cleaning procedures (18 out of 44) and the end of a routine day of work (15 out of 44). On the contrary, in terms of concentration, the level of 5–FU was found to be remarkably higher (7.403 ng/cm^2^) at the end of the cleaning procedures than that (0.036 ng/cm^2^) obtained at the end of nursing personnel shifts. Similarly noteworthy, is the impact of the standard procedures for cleaning of AD residues. Hospital sanitation (HS) personnel assigned to the hospital settings (pharmacies and clinics) who were in charge of the cleaning methods performed their duties using disposable cleaning cloths specific for sanitation services. However, the sampling locations in patient care units showed that AD residues were persistent even after the completion of the cleaning procedures. As an example, 5–FU, at the armrest of the patient treatment chair, obtained a value of 96.315 ng/cm^2^ after the cleaning procedures in 2019. This position was found to be contaminated by the same drug with a level of 47.953 ng/cm^2^ after the nurses had completed their shift.

#### 3.3.2. Pad Samples

The frequency of positive pad samples was 10% in patient care units (Table 3). Of the 282 pad samples (1128 measurements) analyzed, 117 measurements were above the LOD for at least one investigated drug. In particular, the highest value (2.460 ng/cm^2^) of 5–FU was recorded in 2017. This data was obtained by analyzing the sample put on the thorax of a nurse working at outpatient wards. For 5–FU, the positive pad sample (0.502 ng/cm^2^) was in correspondence with the positive wipe samples collected on the armrest of the patient chair in 2019. 

## 4. Discussion

In the pharmacy areas, 263 wipe measurements out of 489 were above the LOD for GEM (54%), 249 for CP (51%), 215 for Pt (44%), and 141 for 5–FU (29%). In the patient care areas, of the 1208 wipe samples taken, 852 were above the LOD for GEM (70%), 824 for Pt (68%), 776 for CP (64%), and 401 for 5–FU (33%).

These results are in accordance with other studies that have shown workplace contamination based on testing hazardous drugs [28]. A 2022 study [23] showed that a measurable concentration of CP was frequently detected on surfaces of different Canadian hospitals. Most front worktops of the BSCs, floors in front of the BSCs in pharmacy areas, and armrests in patient care units were contaminated. Moreover, this study demonstrated that the monitoring program showed a reduction in cyclophosphamide concentration measured on surfaces between 2010 and 2020. Our findings confirm that surface contamination with hazardous drugs persists in the monitored workplaces for all the tested drugs. Despite being the least detected drug in terms of proportion of positive samples, 5–FU had the highest detectable levels on surfaces. There are many factors to consider that explain the spread of 5–FU which include not only aspects, such as workload, work policy and procedures, but also the solubility of drugs: 5–FU is less water-soluble than CP and GEM, therefore, it is more difficult to remove its residual contamination from surfaces in comparison with other compounds.

As far as the proportion of positive pad samples is concerned, the obtained percentages showed different results from those of positive wipe samples. In this study, the low frequency of positive pad samples showed stable trends of contamination not directly comparable to the variability observed for the positive wipe samples. This observation supports the hypothesis that pharmacy technicians operated in full compliance with handling guidelines. Repeated surveys help hospital personnel follow prescribed work procedures very carefully. However, our results showed a different frequency of positive data for pad samples in the pharmacy rather than in the patient care units. The percent of data above the LOD was higher in the pharmacy areas than that in the patient care units for all the monitored drugs with GEM (35%), 5–FU (28%), Pt (14%), and CP (13%). The frequency of positive samples in the patient care units was 20% for GEM, 10% for CP, 7% for 5–FU, and 5% for Pt.

By statistical analysis, we found that, in patients care units and for all compounds, the odds of contamination were always significantly smaller in pad samples than in wipe samples over the 5-year survey. A different pattern was observed in the pharmacy areas, where the odds of contamination were not always significantly different between pad and wipe samples. What is more, we observed for 5–FU in 2019 a significantly higher risk of contamination in pad samples (OR: 3.07; 95% CI: 1.16–8.09; *p* = 0.023). The occurrence of these different outcomes between pharmacy and administration areas is likely due to a larger amount of drugs handled in the pharmacy areas. These findings support results from previous studies [22]. Despite work procedures for the compounding of hazardous drugs being implemented in pharmacies, surface contamination may occur from many sources including broken drug vials, leaking vials, leaking intravenous lines, crushing tablets, or contaminated hands, gloves, and equipment, as reported by Connor et al. [28] in 2017. Tasks of compounding may put technicians at a higher risk of exposure than personnel involved in administering drugs diluted in infusion bags. In addition, these results appear consistent with the higher amount of 5–FU prepared in pharmacy areas.

In order to better understand the risk of exposure in the diverse AD related tasks and define the relationship between surface area contamination and the internal dose in AD engaged workers, biological monitoring may represent a valuable approach. In this study urine data are not reported because a performance-based guidance has been derived to evaluate whether occupational exposure has been adequately controlled. In fact, it may be argued that when surface contamination level is controlled to ng/cm^2^ (90th tiles), the absence of uptake of environmental contaminants would likely be possible. Moreover, because the major route of absorption of AD is dermal, barriers for skin protection such as specific gloves and gowns are expected to limit their uptake [27]. However, it should be noticed that, to confirm the above mentioned statements, further investigations should overcome the limitation of biomonitoring studies due to the analysis of urine that is commonly directed to the quantification of the parent compounds (e.g., CP, GEM, Pt) except for a-fluoro-b-alanine for 5–FU exposure. In particular, the detection of the main metabolites of CP, classified as carcinogenic to humans by the IARC, would be a possible project for future research. Moreover, as biomonitoring results are greatly influenced by the interindividual variability in toxicokinetics, results obtained by the environmental monitoring strategy complete the final risk assessment for hospital personnel involved in the handling of ADs.

There are other factors we considered when conducting wipe sampling. Tests for the four ADs were performed during the pre- and postwork shifts at both pharmacies and patient care wards. In this study, results showed evidence of contamination with ADs after the completion of the cleaning procedures. The composition of the agents embedding cleaning cloths was considered as a possible cause. Residues of hazardous drugs on facility surfaces were found to be higher when using sodium dodecyl sulfate/isopropanol (80:20, v/v) than those found when using 5% sodium hypochlorite. The three top contaminated locations were the pole of infusion bags, trays for drug delivery, and armrests for patient treatment chairs. These findings were comparable with a very recent study by Labrèche et al. [29]. The authors reported that the highest concentrations on surfaces were found when cleaning tasks were carried out by health sanitation personnel. In light of the presence of contamination on surfaces regularly touched by HS personnel, practitioners reviewed the cleaning procedures implemented at the administration areas in 2020, including the use of proper disposable cleaning cloths. In addition, detectable concentrations of ADs on surfaces such as benchtop for drug checking, storage shelves, transfer chambers, floors in front of the BSC, which were not strictly related to the drug compounding and drug administration (e.g., handles, tablet touch-screen, barcode surface) were found. This observation demonstrates that contamination was often spread from highly contaminated areas to other uncontaminated objects present in the workplaces. A possible interpretation of this data is directly related to inadequate cleaning procedures that may spread contaminants. This point was considered important and critical because the possibility that surface contamination is not controlled in the work environment may represent a great risk of exposure and hazard.

Conversely, no action was taken at the BSC workstations in pharmacies because the contamination dropped over time. Pharmacy technicians were well-trained in both preparing and cleaning AD residues. The maximum concentration of 5–FU determined as 9.270 ng/cm^2^ was likely due to an accidental spill occurrence.The concentrations of each drug over the 5-year survey are described by the percentiles of data distribution. CP had the 90th percentiles below the reported limit of 0.1 ng/cm^2^ at both pharmacy areas and patient care units [15]. The CP 90th percentile was 0.026 ng/cm^2^ in pharmacies and 0.070 ng/cm^2^ in patient care units. This is also the case in other studies. For example, Chauchat et al. [21] determined the 90th percentile of CP contamination on all pharmacy surfaces as 0.029 ng/cm^2^ and all clinic surfaces as 0.032 ng/cm^2^. Jeronimo et al. [22] determined the 90th percentile of CP contamination on all pharmacy surfaces as 0.031 ng/cm^2^ and all clinic surfaces as 0.018 ng/cm^2^.

In our study, for pharmacy areas, a technical guidance based on the percentile of 1956 wipe measurements was suggested. As a result, the 90th percentile was chosen based on the most detected compound (9.270 ng/cm^2^). According to the recent literature, we propose this guidance for comparison reasons in order to identify the best work policy of drug compounding under different work practices [15,23]. The 90th percentile of 5–FU contamination was 0.346 ng/cm^2^ over the 5-year survey. More in detail, the 5–FU percentiles varied over the monitoring program studies (Figure 3). The maximum value was between 2016 and 2019. Then, a peak improvement occurred in 2020 (0.087 ng/cm^2^, 90th percentile) followed by another contamination increase in 2021 (0.496 ng/cm^2^).

This trend of contamination was similar in the patient care units. For this reason, the 5–FU, 90th percentile resulted as 0.443 ng/cm^2^ (2016–2021).

Trends of 5–FU surface contamination and its variability were likely due to a continuous increasing amount of the drug prepared by pharmacy technicians over time. In addition, its poor solubility in aqueous-based solutions was deemed as a possible and critical point in performing cleaning procedures.

The present Italian survey demonstrated that surface contamination with hazardous drugs has been decreasing over time, although detectable levels of ADs are still present in the investigated workplaces. As a result, the 90th percentile from the obtained data is higher than that reported by Germany and other European studies [15,18]. This observation has led practitioners to apply the 90th percentile as a technical threshold limit in order to make pharmacy technicians and nursing personnel aware of risks with the final aim of understanding when contamination is adequately controlled.

## 5. Conclusions

In the present study, surface contamination was discussed. Residues of hazardous drugs on facility surfaces, including pharmacies and patient care areas, showed detectable levels of ADs. This 5-year survey provided evidence that the proportion of positive wipe samples is greater in patient care units than in pharmacy areas and the concentrations of ADs based on the 90th percentile data distribution was less than 0.346 ng/cm^2^ in pharmacies and less than 0.443 ng/cm^2^ in patient care areas. AD concentrations of contamination are higher than those reported for facilities in other European countries yet trends of contamination in Italy have been shown to decrease over time. This study, based on the stored data, has suggested a benchmark comparable with other studies to evaluate how much the residual presence of ADs is still an issue in workplaces and may represent a health hazard. This observation corroborates the body of knowledge in this area. Repeated monitoring programs displayed the importance of raising the awareness of risk for hospital personnel involved in the handling of ADs.

In the near future, it is hoped that more hospital centers can take part in this study in order to obtain a national and, therefore, more complete overview regarding technical limit values for ADs comparable with those reported in the literature. Moreover, these limits may be used in the field for protecting the health and safety of hospital personnel.

## Figures and Tables

**Figure 1 ijerph-19-08601-f001:**
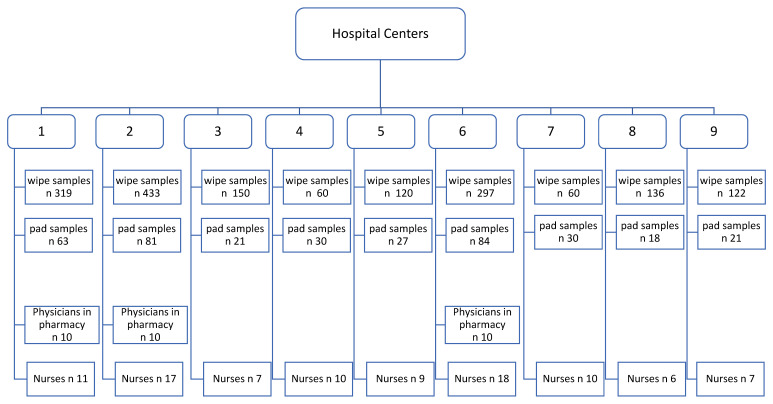
Schematic presentation of the study design.

**Figure 2 ijerph-19-08601-f002:**
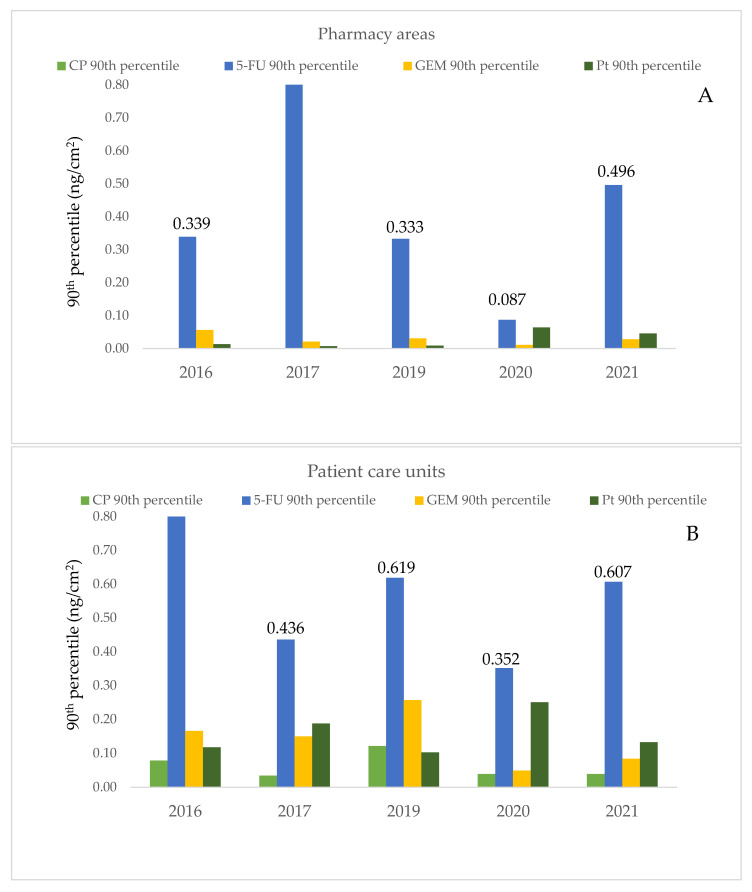
Trends of contamination in pharmacy areas (**A**) and patient care units (**B**) based on the 90th percentile of data distribution for each drug (CP=cyclophosphamide, 5-FU = 5-fluorouracil, GEM = gemcitabine, Pt = platinum compounds) and over a 5-year monitoring survey.

**Figure 3 ijerph-19-08601-f003:**
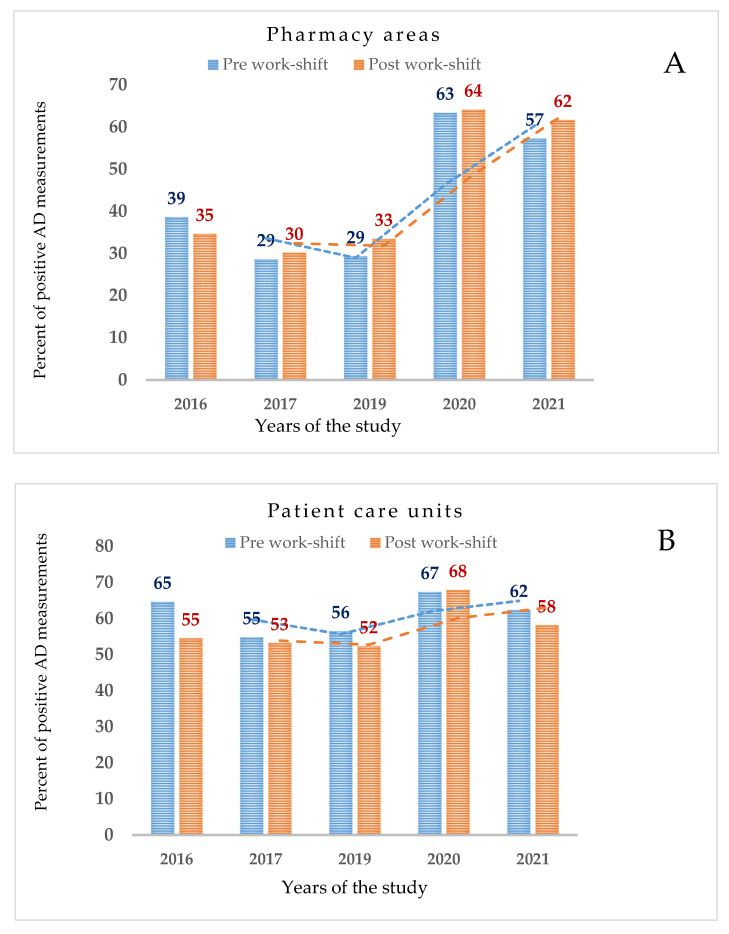
In panel (**A**) (pharmacy areas) and in panel (**B**) (patient care units) percent of positive wipe samples in relation to the pre- and postwork shift over the 5-year survey are depicted.

**Table 1 ijerph-19-08601-t001:** Description of participating centers: pharmacy and patient care areas.

Years of the Study (2016–2021)			Hospital Center							
	Antineoplastic Drugs	1	2	3	4	5	6	7	8	9
**Qualitative Classification of Center size**		Large	Large	Large	Medium	Small	Medium	Small	Small	Large
**Total number of beds**		545	567	223	77	32	76	42	48	244
**Presence of Pharmacy Unit**		yes	Yes	no	no	no	yes	no	no	no
**Number of Technicians Involved in Compounding**		7	6				8			
**Presence of Patient Care Units**		yes	Yes	yes	yes	yes	yes	yes	yes	yes
**Presence of Inpatient Beds**		yes	Yes				yes			
**Presence of Outpatient Seats**		yes	Yes	yes	yes	yes	yes	yes	yes	yes
**Average Amount (mg) of each Drug Prepared on the Sampling Day**	CP ^1^	18,184	16,160				6840			
	5–FU ^2^	155,953	130,934				46,334			
	GEM ^3^	66,332	58,377				18,858			
	Pt ^4^	9358	8053				4688			
**Number of Technicians Involved in Administration**		46	79	17	9	10	58	6	2	13
**Average Number of Patients Treated per Day**		49	66	30	13	24	67	36	4	23
**Closed System Transfer Device Used in the Last 6 Years**		yes	Yes	yes	yes	yes	yes	yes	yes	yes

^1^ CP = cyclophosphamide, ^2^ 5-FU = 5-fluorouracil, ^3^ GEM = gemcitabine, ^4^ Pt= platinum compounds.

**Table 2 ijerph-19-08601-t002:** Frequency of contamination for each drug throughout the years of study in wipe samples.

		Percent of Data Above LOD %				Total Number of Data (Positive Data)	
	Years of Study	CP ^1^	5–FU ^2^	GEM ^3^	Pt ^4^		% > LOD ^5^
**Pharmacy Areas,** **n = 489**	2016	42	22	36	47	352 (129)	37
	2017	24	20	35	39	384 (113)	29
	2019	44	24	32	26	420 (132)	31
	2020	75	54	83	43	420 (268)	64
	2021	67	22	80	68	380 (226)	59
	2016–2021	51	29	54	44		
	Subtotal of data (positive data)						% > LOD ^5^
	2016–2021	489 (249)	489 (141)	489 (263)	489 (215)	1956 (868)	44
		Percent of data above LOD %				Total number of data (positive data)	
	Years of study	CP^1^	5–FU ^2^	GEM ^3^	Pt ^4^		% > LOD ^5^
**Patient Care Units,** **n = 1208**	2016	68	25	67	78	928 (553)	60
	2017	44	31	69	72	944 (510)	54
	2019	64	42	62	50	988 (538)	54
	2020	77	55	78	62	844 (571)	68
	2021	68	19	77	77	1128 (681)	60
	2016–2021	64	33	77	68		
		Subtotal of data (positive data)					% > LOD ^5^
	2016–2021	1208 (776)	1208 (401)	1208 (852)	1208 (824)	4832 (2853)	59

^1^ CP = cyclophosphamide, ^2^ 5-FU = 5-fluorouracil, ^3^ GEM = gemcitabine, ^4^ Pt = platinum compounds, ^5^ LOD = lower limit of detection.

**Table 3 ijerph-19-08601-t003:** Frequency of contamination for each drug throughout the years of study in pad samples.

		Percent of Data above LOD %				Total Number of Data (Positive Data)	
	Years of Study	CP ^1^	5–FU ^2^	GEM ^3^	Pt ^4^		% > LOD ^5^
**Pharmacy Areas,** **n = 93**	2016	6	28	28	22	72 (15)	21
	2017	11	33	39	6	72 (16)	22
	2019	38	48	33	19	84 (29)	35
	2020	11	11	33	6	72 /11)	15
	2021	<LOD	28	56	22	72 (19)	26
	2016–2021	14	30	38	15		
		Subtotal data (positive data)					% > LOD
	2016–2021	93 (13)	93 (28)	93 (35)	93 (14)	372 (90)	24
		Percent of Data above LOD %				Total Number of Data (Positive data)	
	Years of study	CP ^1^	5–FU ^2^	GEM ^3^	Pt ^4^		% > LOD ^5^
**Patient Care Units,** **n = 282**	2016	16	7	17	7	300 (35)	12
	2017	7	10	25	9	276 (35)	13
	2019	4	9	18	<LOD	180 (14)	8
	2020	12	6	36	3	132 (19)	14
	2021	7	3	10	3	240 (14)	6
	2016–2021	10	7	20	5		
		Subtotal data (positive data)					% > LOD ^5^
	2016–2021	282(27)	282 (20)	282 (56)	282 (14)	1128 (117)	10

^1^ CP = cyclophosphamide, ^2^ 5-FU = 5-fluorouracil, ^3^ GEM = gemcitabine, ^4^ Pt = platinum compounds, ^5^ LOD = limit of detection.

**Table 4 ijerph-19-08601-t004:** Surface contamination reported as the 75th percentile, the 90th percentile, and the maximum value for each drug over the 5-year survey in wipe samples.

Surface Concentration for Each Drug in Pharmacy Areas													
		CP ^1^ (ng/cm^2^)			5–FU ^2^ (ng/cm^2^)			GEM ^3^ (ng/cm^2^)			Pt ^4^ (ng/cm^2^)		
	Sampling Location	75th	90th	Max	75th	90th	Max	75th	90th	Max	75th	90th	Max
	BSC work surface * n = 672	0.001	0.002	0.010	0.278	1.375	9.270	0.005	0.014	0.677	0.006	0.017	0.177
	Checking counter n=152	0.002	0.004	0.012	0.013	0.019	0.022	0.002	0.016	0.019	0.002	0.005	0.008
**Handles**	Passthrough handle (inside cleanroom) n = 104	0.019	0.118	0.300	0.553	1.097	1.460	0.062	0.100	2.700	0.075	0.118	0.431
	Door handle (inside) n = 120	0.007	0.007	0.010	0.333	0.333	0.333	0.006	0.007	0.007	0.007	0.007	0.018
	Fridge handle (inside) n = 48	0.021	0.031	0.050	0.508	0.742	0.898	0.008	0.013	0.017	0.015	0.019	0.022
**Floors**	Floor in front of BSC n = 160	0.056	0.167	0.423	0.065	0.125	5.974	0.028	0.050	0.107	0.004	0.066	0.129
	Floor in cleanroom n = 96	0.011	0.014	0.022	0.030	0.210	0.250	0.005	0.020	0.045	0.005	0.005	0.005
	Floor in front of cleanroom n = 32	<LOD	<LOD	<LOD	0.081	0.095	0.104	<LOD	<LOD	<LOD	0.001	0.001	0.001
	Floor in anteroom n = 120	0.028	0.039	0.092	0.062	0.083	0.083	0.009	0.027	0.038	0.006	0.020	0.062
	Floor in front pass-through n = 24	0.001	0.001	0.001	<LOD	<LOD	<LOD	0.012	0.018	0.022	0.002	0.002	0.002
**Trays**	Trays used for drug delivery n = 152	0.020	0.032	0.045	0.084	0.289	1.735	0.002	0.025	0.193	0.012	0.039	0.043
	Trays/countertops used at the nurse’s station for documentation n = 96	0.003	0.007	0.021	0.012	0.026	0.084	0.001	0.003	0.010	0.0001	0.001	0.007
	Infusion bag surface n = 44	0.001	0.001	0.001	0.052	0.132	0.132	0.001	0.001	0.001	0.001	0.001	0.001
	Mouse PC n = 32	0.007	0.008	0.008	0.160	0.200	0.227	0.002	0.002	0.002	0.002	0.002	0.002
	Barcode surface n = 32	0.005	0.015	0.080	0.088	0.460	0.550	0.022	0.057	0.708	0.004	0.052	0.523
	Subtotal n = 489 (1956 data)	0.007	0.026	0.423	0.107	0.346	9.270	0.007	0.028	2.700	0.006	0.026	0.523

* BSC work surface (i.e., area sampled inside BSC on the left, right, and central part of the BSC working top). ^1^ CP = cyclophosphamide, ^2^ 5-FU = 5-fluorouracil, ^3^ GEM = gemcitabine, ^4^ Pt = platinum compounds.

**Table 5 ijerph-19-08601-t005:** Surface contamination reported as the 75th percentile, the 90th percentile, and the maximum value for each drug over the 5-year survey in wipe samples.

Surface Concentration for Each Drug in Patient Care Units													
		CP ^1^ (ng/cm^2^)			5–FU ^2^ (ng/cm^2^)			GEM ^3^ (ng/cm^2^)			Pt ^4^ (ng/cm^2^)		
	Sampling Location	75th	90th	Max	75th	90th	Max	75th	90th	Max	75th	90th	Max
**Counters**	Countertop used for validation n = 76	<LOD	0.001	0.001	0.013	0.029	0.040	0.004	0.015	0.020	0.001	0.005	0.011
	Countertop used for deposition of ready-to-use bags n = 664	0.002	0.007	0.201	0.066	0.260	112.500	0.004	0.013	0.117	0.003	0.005	0.058
	Bedside table used by patients n = 256	0.005	0.021	0.097	0.198	0.356	0.440	0.017	0.038	0.296	0.007	0.013	0.039
	Pole for infusion bags n = 704	0.038	0.742	27.023	0.532	1.464	5.560	0.049	0.184	4.753	0.020	0.071	0.986
	Bed bell surface used by patient n = 16	0.233	0.256	0.271	<LOD	<LOD	>LOD	0.020	0.021	0.022	0.001	0.001	0.001
	Armrest of patient treatment chair n = 104	0.024	0.109	0.187	0.319	0.892	2.678	0.004	0.077	0.351	0.027	0.943	1.411
	Touch-screen of the perfusion pump n = 244	0.011	0.018	0.125	0.099	0.243	2.269	0.027	0.087	0.543	0.006	0.031	0.173
	Trays used for drug delivery n = 88	0.003	0.036	0.621	5.560	6.666	7.403	0.001	0.002	0.013	< LOD	< LOD	< LOD
	Tablet touch-screen n = 144	0.003	0.004	0.004	0.134	0.327	0.574	0.008	0.023	0.069	0.003	0.004	0.005
	Barcode surface n = 324	0.014	0.080	0.257	0.119	0.634	6.825	0.013	0.036	0.430	0.001	0.002	0.004
	Mouse PC n = 52	0.056	0.110	0.146	0.672	1.533	2.108	0.046	0.049	0.051	0.002	0.002	0.002
**Floors**	Floor in front of pole n = 580	0.012	0.056	0.924	0.141	0.264	3.016	0.028	0.085	0.424	0.025	0.062	0.246
	Floor in restroom n = 496	0.017	0.068	1.208	0.202	1.463	9.154	0.239	0.721	12.998	0.314	1.153	4.913
	Floor in front of restroom n = 168	0.018	0.099	0.634	0.135	0.273	1.454	0.088	0.162	1.345	0.196	0.454	1.948
	Floor in patient room n = 224	0.004	0.007	0.015	0.096	0.200	0.361	0.017	0.047	0.627	0.018	0.040	0.118
	Floor in nursing room n = 172	0.003	0.008	0.054	0.029	0.137	0.348	0.007	0.012	0.103	0.004	0.033	0.066
	Floor in storage room n = 188	0.003	0.011	0.205	0.247	0.300	0.322	0.037	0.092	1.107	0.040	0.082	0.264
**Handles**	Handle in restroom (inside) n = 152	0.095	0.280	9.908	0.417	0.417	0.417	0.039	0.214	0.508	0.017	0.037	0.206
	Fridge handle (inside) at the nurse station n = 88	0.008	0.017	0.028	0.088	0.088	0.088	0.008	0.019	0.023	0.002	0.005	0.006
	Toilet surface in restroom n = 48	0.005	0.006	0.006	0.031	0.042	0.050	0.016	0.323	0.536	0.096	0.107	0.121
	Subtotal n = 1208 (4832 data)	0.012	0.070	27.023	0.191	0.443	112.500	0.034	0.146	12.998	0.033	0.141	4.913

^1^ CP = cyclophosphamide, ^2^ 5-FU = 5-fluorouracil, ^3^ GEM = gemcitabine, ^4^ Pt = platinum compounds.

## Data Availability

The data that support the findings of this study are available on request from the corresponding author.

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
