# Peer review of "Occupational Exposure Assessment to Antineoplastic Drugs in Nine Italian Hospital Centers over a 5-Year Survey Program"

_ijerph, 2022, doi:10.3390/ijerph19148601_

Round 1

Reviewer 1 Report

This paper describes surface contamination and dermal exposure in Italian hospitals for a number of antineoplastic drugs over time. Surface contamination is a useful method to perform exposure surveillance to set priorities, raising awareness and consider mitigating measures. There are some concerns opportunities for improvement. The manuscript is incomplete with respect to characteristics of the nine hospitals and the results of urine analysis and the discussion of these results.

Major

1.       The title is not clear and could be misread/interpreted easily due to the words ‘drug levels’ which is not sufficiently specific. Does it also relate to internal exposures? It is suggested to change the title to refer to the main type of data that were presented (e.g. surface contamination and skin contact measurements).

2.       When describing the study design it is not sufficiently clear that the main focus is on surface wipe sampling and additional pad samples (on the person). Can the authors provide a table with the characteristics of the nine hospitals (e.g. university vs. peripheral hospitals, number of beds, number of workers per job title and/or department, etc.). The few lines (L108-110 are not sufficient). It would also help to declare the sample collection effort by hospital for the different sample types and how they were distributed over time and by hospital.

3.       The method of surface wipe sampling is not sufficiently clear. Did the researchers clean the surface assigned as sample location (and using the tape to mark the surface area)? If not the t=0 situation is not clearly defined. It should be explained that the results of the surface wipe data may be undefined historic contamination. This is different from the situation of a more defined time-window and then the t=0 should be described (start of a workweek or start of a shift?). Another question regarding surface wipes is how the defined surface area and calculations to ug/cm2 were done for small non-planar objects like e.g. infusion bags, PC mouse, door handles, etc. With regards to the pads the authors should declare the metric dimensions (size and shape). Are these pads worn under clothes or not.

4.       Regarding the analysis of wipes and pads the authors should share information on the recovery rate for both sample extraction methods for each antineoplastic drug. It is not sufficient to (only) refer to an earlier published method here! If recovery was lower than 100%, did the authors correct for that or not? The authors should also declare the LOQ and LOD for each drug. The same questions should be answered for the analysis of urine for parents and metabolites.

5.       Regarding the results of urine analysis the one line in the introduction (L107-108) that no antineoplastic drugs were detected is not sufficient. This should be addressed in both results and discussion section with an explanation of why the authors did not find any samples with detects.

6.       In the discussion it is important to provide some more interpretation the surface-to-surface transfer e.g. due to the cleaning. Does cleaning process lead to a reduction of surface contamination or do the authors also consider the cleaning process contribute to the problem of dispersion existing localized and high contamination to other (initially) uncontaminated surfaces?

7.       Do the authors understand the change of the frequency of detects (%>LOD) in the pharmacy areas where an increase was observed from around 30-40% in 2016-2019 to around 60 in 2020-2021. How does this translate to %>LOD for pad samples where this change is not observed?

8.       The authors should discuss how to interpret skin contact for the different drugs studied in terms of health risk. E.g. which of the drugs can be skin absorbed and to what extent? There are also differences in the toxic potency? Which is the most critical drug in terms of exposure and hazard?

Minor

L272  ‘detail’ instead of ‘details’

L281 ‘in correspondence with the pole were’ is unclear, suggest to rephrase

L352-353 Clarify/specify the definition of the reference situation for these ORs

L382 Suggest to add a literature reference for the limit of 0.1 ng/cm2

Author Response

Major

  1. The title is not clear and could be misread/interpreted easily due to the words ‘drug levels’ which is not sufficiently specific. Does it also relate to internal exposures? It is suggested to change the title to refer to the main type of data that were presented (e.g. surface contamination and skin contact measurements).

We agree with this observation. The title has been rephrased in order to clarify the meaning of “drug levels”. “Surface contamination and skin contact measurements in nine Italian hospital centers over a 5-year survey program”

  1. When describing the study design it is not sufficiently clear that the main focus is on surface wipe sampling and additional pad samples (on the person). Can the authors provide a table with the characteristics of the nine hospitals (e.g. university vs. peripheral hospitals, number of beds, number of workers per job title and/or department, etc.). The few lines (L108-110 are not sufficient). It would also help to declare the sample collection effort by hospital for the different sample types and how they were distributed over time and by hospital.

We have accepted this point and more details have been reported in Table 1. The main characteristics of the investigated hospital centers are reported in this table. The number of samples for the different sample types and their distribution over time by each hospital have also been reported. Study design paragraph has been improved and includes a new Table. Sampling techniques paragraph has been added.

  1. The method of surface wipe sampling is not sufficiently clear. Did the researchers clean the surface assigned as sample location (and using the tape to mark the surface area)? If not the t=0 situation is not clearly defined. It should be explained that the results of the surface wipe data may be undefined historic contamination. This is different from the situation of a more defined time-window and then the t=0 should be described (start of a workweek or start of a shift?). Another question regarding surface wipes is how the defined surface area and calculations to ug/cm2 were done for small non-planar objects like e.g. infusion bags, PC mouse, door handles, etc. With regards to the pads the authors should declare the metric dimensions (size and shape). Are these pads worn under clothes or not.

Thank you for your observation because it has given us the opportunity to provide an explanation of the t=0 situation. In the study design paragraph details regarding the monitoring strategy have been described. To provide a schematic presentation of the whole study, Figure 1 has been added. The description of the sample distribution over time by hospital and the hospital personnel described per job title has been included.

  1. Regarding the analysis of wipes and pads the authors should share information on the recovery rate for both sample extraction methods for each antineoplastic drug. It is not sufficient to (only) refer to an earlier published method here! If recovery was lower than 100%, did the authors correct for that or not? The authors should also declare the LOQ and LOD for each drug. The same questions should be answered for the analysis of urine for parents and metabolites.

The results were not corrected for the recovery values. As the recovery from wipe sampling varied between 80% and 95 % for linoleum and stainless steel, we considered that it would not have a great impact on the exposure estimates. To our knowledge, results reported in literature are not corrected for recovery, therefore, for comparison, we used raw data [15, 16].

We accepted this point and LOQ values have been reported along with the reported LOD values in the new version of the analytical methods paragraph.

  1. Regarding the results of urine analysis the one line in the introduction (L107-108) that no antineoplastic drugs were detected is not sufficient. This should be addressed in both results and discussion section with an explanation of why the authors did not find any samples with detects.

We have accepted this point. We have discussed biomonitoring results in the field of AD exposure and some comments have been reported in the discussion paragraph.

  1. In the discussion it is important to provide some more interpretation the surface-to-surface transfer e.g. due to the cleaning. Does cleaning process lead to a reduction of surface contamination or do the authors also consider the cleaning process contribute to the problem of dispersion existing localized and high contamination to other (initially) uncontaminated surfaces? We have improved this critical point and the issue regarding the cleaning procedures for AD still present in the work environment has been reported with more interpretation.

  1. Do the authors understand the change of the frequency of detects (%>LOD) in the pharmacy areas where an increase was observed from around 30-40% in 2016-2019 to around 60 in 2020-2021. How does this translate to %>LOD for pad samples where this change is not observed?

We thank the reviewer for this point. Additional interpretation has been reported in the discussion paragraph.

  1. The authors should discuss how to interpret skin contact for the different drugs studied in terms of health risk. E.g. which of the drugs can be skin absorbed and to what extent? There are also differences in the toxic potency? Which is the most critical drug in terms of exposure and hazard? We have discussed this point in correspondence with the previous point 5 regarding the biomonitoring results and possible uptake of AD in the field of the exposure to these active compounds.

Minor

L272  ‘detail’ instead of ‘details’ It was checked and corrected

L281 ‘in correspondence with the pole were’ is unclear, suggest to rephrase It was rephrased.

L352-353 Clarify/specify the definition of the reference situation for these ORs

An explanation of ORs has been provided in the statistical analysis paragraph

L382 Suggest to add a literature reference for the limit of 0.1 ng/cm2

We have accepted this suggestion. The reference has been added.

Reviewer 2 Report

Non-specialists will struggle to read this paper on an issue of much interest to health care professionals and the occupational cancer field.

"Pad" samples need to be better defined when they are first mentioned (3:104).  I surmise they are integrated dosimeters of skin exposure, but a little more background is needed.

It's a wasted opportunity to not show a few figures with cumulative distributions, given the frequent references to %-iles throughout the manuscript.  These data are currently in a couple of tables.  But I believe the presentation would be enhanced with a few figures, instead or in supplementing the great quantity of data.  The two figures, which are not especially "meaty", might work better in a table, but that is not essential.

Paragraph 1 on page 10 is too, too dense with data. Break it into at least two, maybe three, paragraphs.

11: I suggest "measured, characterized statistically, and discussed."

Numerous English fixes below

data points 1:25, 2:91

organizational 2: 57-8

profileS 2:72

"More in details" needs fixing 8:272

removing procedures 9:300 is awkward

10:373 in charge of is backwards.   The people are in charge of the methods

11:49  been shown

11:420:  based on accumulated, stored data

11:426 "vision" usually denotes a big picture ideal toward which leaders are striving, not a snapshot of the state of a field

7:230  was experimented needs to be revised.

Author Response

Comments and Suggestions for Authors

Non-specialists will struggle to read this paper on an issue of much interest to health care professionals and the occupational cancer field.

"Pad" samples need to be better defined when they are first mentioned (3:104).  I surmise they are integrated dosimeters of skin exposure, but a little more background is needed.

We thank the reviewer for this important comment. An extra paragraph regarding the sampling technique has been added.

It's a wasted opportunity to not show a few figures with cumulative distributions, given the frequent references to %-iles throughout the manuscript.  These data are currently in a couple of tables.  But I believe the presentation would be enhanced with a few figures, instead or in supplementing the great quantity of data.  The two figures, which are not especially "meaty", might work better in a table, but that is not essential

We have accepted this point. Figure 2 has been added

Paragraph 1 on page 10 is too, too dense with data. Break it into at least two, maybe three, paragraphs. We have agreed with this observation, we have changed this sentence accordingly.

11: I suggest "measured, characterized statistically, and discussed."

Numerous English fixes below

We have accepted all English fixes pointed out by reviewer 2.

data points 1:25, 2:91accepted

organizational 2: 57-8 accepted

It has been rephrased.

profileS 2:72

"More in details" needs fixing 8:272 It was checked and corrected

removing procedures 9:300 is awkward

it was changed.

10:373 in charge of is backwards.   The people are in charge of the methods

We have reviewed this phrase

11:49  been shown

11:420:  based on accumulated, stored data

accepted

11:426 "vision" usually denotes a big picture ideal toward which leaders are striving, not a snapshot of the state of a field

We have changed this term to a more adequate one for the context.

7:230  was experimented needs to be revised.

We have revised this point

Reviewer 3 Report

This study addresses an important issue such as occupational exposure to antineoplastic drugs (ADs) which is still the subject of scientific debate today, particularly in Italy where no investigations to define surface contamination thresholds have been recently developed according to the lower limit of detection values (LODs) of the most recently validated analytical methods. I believe that this work represents a starting point for future analyzes in which there will be the involvement of more hospitals from different Italian regions aimed at obtaining a national and deeper vision regarding technical limit values for ADs comparable wich may be used in the field for protecting health and safety of hospital personnel.

For these reasons I believe that the present study has a high scientific soundness. Also the research design is appropriate and the methods are adequately described; the results are clearly presented and well discussed.

Considering the above strenghts, my overall recommendation is to accept the manuscript in present form

Author Response

Thank you for your kind consideration. We agree with your suggestion regarding the importance of involving more Italian hospitals in the near future studies.